# How Do Gene Expression Patterns Change in Response to Osmotic Stresses in Kuruma Shrimp (*Marsupenaeus japonicus*)?

**Yuquan Li** [†], **Zhihao Zhang** [†], **Zhongkai Wang, Zhitong Deng, Ruiyang Zhao, Jinfeng Sun, Pengyuan Hao, Long Zhang, Xiaofan Wang, Fei Liu, Renjie Wang and Yanting Cui** *

School of Marine Science and Engineering, Qingdao Agricultural University, Qingdao 266109, China
* Correspondence: cuiyanting1114@qau.edu.cn
† These authors contributed equally to this work.

**Abstract:** Euryhaline crustaceans cope with external salinity changes by mechanisms of osmoregulation. In the current study, we first cloned and confirmed the ORF sequences of the ion-transportation-related genes $Na^+/K^+$-ATPase $\alpha$ subunit (NKA$\alpha$), cytoplasmic carbonic anhydrase (CAc), and V-type $H^+$-ATPase G subunit (VHA-G), and water channels of aquaporins (AQP3, AQP4, and AQP11) from kuruma shrimp (*Marsupenaeus japonicus*). Further tissue expression patterns showed a higher expression of *MjAQP4*, *MjCAc*, *MjNKAα*, and *MjVHA-G* in the gills, as well as a higher expression of *MjAQP3* and *MjAQP11* in the intestine and muscle, respectively. Then, qPCR analysis was used to assess the mRNA expression levels of those osmoregulatory genes in both post-larvae and adult shrimp when they were exposed to acute salinity stress or salinity acclimation. The results revealed significantly decreased expression levels of *MjAQP3*, *MjAQP11*, *MjNKAα*, and *MjCAc*, and higher expression levels of *MjAQP4* and *MjVHA-G* when the post-larvae shrimp were directly subjected to 10‰ or 50‰ salinity. Moreover, similar expression patterns were also observed in the post-larvae shrimp during the accommodation to 10‰ or 50‰ salinity. As to the adult shrimp, significantly higher expression levels of those genes were observed in the gills after exposure to 10‰ salinity, whereas only the expression levels of *MjAQP3*, *MjAQP11*, and *MjNKAα* were up-regulated in the gills at 40‰ salinity. In contrast, the expression of *MjVHA-G* was significantly decreased at 40‰ salinity. Finally, during the acclimation to 10‰ salinity, the expression levels of *MjAQP3*, *MjAQP11*, and *MjNKAα* were also significantly elevated, while the expression of *MjCAc* was significantly decreased in the gills. In addition, the expression levels of *MjAQP3*, *MjAQP4*, *MjCAc*, and *MjVHA-G* were significantly decreased in the gills during the acclimation to 55‰ salinity. The findings of the study suggest that the examined genes are critical for the adaptation of aquatic crustaceans to changing environmental salinity. Our study lays as the foundation for further research on osmoregulation mechanisms in *M. japonicus*.

**Keywords:** salinity; $Na^+/K^+$-ATPase; carbonic anhydrase; V-type $H^+$-ATPase; aquaporin

## 1. Introduction

Salinity is one of the major environmental factors that exert crucial effects on aquatic organisms. Variation in salinity can impose osmotic stress, alter osmolyte concentrations, and affect the presence, physiology, abundance, and distribution of aquatic animals [1]. Osmoregulation is believed to be the primary physiological function that aquatic animals use to maintain the ionic and water balance between the body fluid and the environment [2,3]. The widespread distribution of crustaceans in marine, brackish, and freshwater environments demonstrates an extraordinary capacity to adapt to variable salinities, which benefits from their powerful osmotic regulation abilities [4].

Researchers have discovered the physiological and morphological characteristics of crustaceans that facilitate their adaptation to a wide range of salinities [5,6]. The gill is considered the primary organ for osmoregulation and the epithelium of the gill contains

specialized cells responsible for regulating osmotic pressure, controlling gas exchange, and maintaining an acid–base balance [7]. Moreover, crustaceans can respond to environmental salinity changes through the expression of osmoregulatory genes involved in active ion transport and exchange, the tolerance of osmotic stress, water channel maintenance, and cell volume control [8]. Ion transport and exchange are mostly attributed to the activity of the $Na^+/K^+$-ATPase (NKA), V-type $H^+$-ATPase (VHA), and carbonic anhydrase (CA) in the gills of crustaceans [2,3,9]. NKA has been the most extensively examined gene in osmoregulatory studies of euryhaline animals because it is a critical enzyme responsible for establishing electrochemical gradients across the cell membrane in the gills [10]. VHA actively participates in osmoregulatory ion uptake across the gill epithelium of freshwater-tolerant crustaceans and is also responsible for acid–base balance and nitrogen excretion [2,9,11,12], whereas CA rapidly converts carbon dioxide to bicarbonate and carbonate, which affects the acid–base balance, ion transport, and gas exchange at both the tissue and cellular levels [13,14]. Apart from these, aquaporins (AQPs) are small transmembrane channel proteins that mainly facilitate the transport of water and small solutes through cellular membranes [15,16]. AQPs are classified into classical aquaporins, aquaglyceroporins, and unorthodox aquaporins [17,18]. Moreover, the roles of AQPs in osmoregulation have been indicated by their transcriptional regulation upon salinity stress in crustaceans in the last decade [19–24]. Recently, the AQP family from Pacific white shrimp (*Litopenaeus vannamei*) has been characterized and their expression patterns in response to salinity changes have been clarified [25,26].

The kuruma shrimp (*Marsupenaeus japonicus*) is one of the major shrimp species with high economic value that is mainly cultivated in China, accounting for more than 5% of the total shrimp production [27]. Nevertheless, due to the influence of estuaries, seasonal ocean currents, and other factors, changes in salinity are inevitable during aquaculture, which results in large economic losses in the industry. On the other hand, the distribution of salinity in aquatic waters in China is not uniform. There are large areas of inland saline waters and coastal high-salinity waters in China [28–30], wherein most of them have been in an abandoned state for a long time [31]. As *M. japonicus* is a species capable of tolerating a wide range of salinity [32], it can be developed as a promising aquaculture species in these waters to expand the space for aquaculture and increase the incomes of farmers. Thus, studies of the salinity adaptation mechanism of *M. japonicus* could provide new strategies for developing and optimizing successful shrimp farming in variable salinity conditions.

Up to now, few studies have addressed how the osmoregulatory genes respond to salinity challenges in *M. japonicus*. Therefore, the current study aimed to elucidate the potential roles of ion-transportation-related genes (*NKA*, *CA*, and *VHA*) and water channel genes (*AQPs*) with respect to variable salinity. These genes were first identified and characterized from the transcriptome of *M. japonicus.* Then, real-time quantitative PCR (qPCR) analysis was performed to clarify their tissue distributions and expression patterns in response to acute salinity challenges or salinity acclimation at two developmental stages. The investigation of their expression profiles can provide insights into not only the osmoregulation mechanisms of crustaceans but also into the cultivation of *M. japonicus* in the inland saline waters and coastal high-salinity waters of China.

## 2. Materials and Methods

### 2.1. Experimental Animals

The post-larvae shrimp (P5) were obtained from a farm in Weifang city, Shandong province, China. Shrimp were cultured in PVC buckets (30 L) containing 20 L of aerated natural seawater (salinity 30‰, pH 8.0, and 25 ± 0.5 °C) with a 30% daily exchange rate and fed *Artemia* four times a day (8:00, 13:00, 18:00, and 23:00). The following salinity stress experiments of post-larvae shrimp were also conducted at this farm.

Adult shrimp (average weight: 10.0 ± 0.5 g; body length: 11.2 ± 0.4 cm) were reared in open ponds in Qingdao city, Shandong province, China. The temperature of the ponds was 18 ± 1 °C and the salinity was 25‰. Then, the shrimp were transferred to the School of

Marine Science and Engineering, Qingdao Agricultural University. They were acclimated in PVC buckets (300 L) and fed three times per day (9:00, 15:00, and 21:00; 5% of shrimp body weight per feeding) with a commercial pelleted feed (CP Group Feed, Qingdao, China). The temperature and salinity of seawater were re-adjusted to that of the shrimp's original habitat in ponds. The feces and uneaten feed were removed daily with a siphon tube. Dissolved oxygen was kept above 6.0 mg/L.

All handling of this study was performed in strict accordance with the guidelines of the Animal Experimentation Ethics Committee of Qingdao Agricultural University, which also approved the protocol.

### 2.2. Effects of Acute Salinity Changes on Post-Larvae Shrimp

Artificial high-salinity (35‰, 40‰, 45‰, 50‰, and 55‰) seawater was prepared by adding sea salt to natural seawater (30‰), while low-salinity (25‰, 20‰, 15‰, 10‰, and 5‰) seawater was made by the addition of pre-aerated freshwater to natural seawater. A water quality detector (YSI incorporated, USA) was used to measure and monitor the salinity of seawater until it reached the target salinity. The pH and water temperature of artificial seawater were consistent with those of natural seawater. The post-larvae shrimp were directly placed from seawater of 30‰ salinity to seawater of high-salinity with ascending gradient or low-salinity with descending gradient. Beakers containing 500 mL of seawater with 10 shrimp each were prepared in triplicate for each salinity concentration. The number of surviving individuals in each group was counted and the dead ones were removed at 12 h and 24 h (hours) after the salinity challenges.

Based on the survival rates of shrimp from the above experiment, 10‰ and 50‰ salinity concentrations were set as the treatment salinity. Forty-five post-larvae shrimp (fifteen shrimp/bucket × three buckets) reared at 30‰ salinity were flash-frozen in liquid nitrogen and set as the control group (0 h) for this experiment. Then, shrimp were separately transferred from seawater of 30‰ salinity to 10‰ or 50‰ salinity and each treatment was replicated thrice (40 shrimp per replicate). Forty-five shrimp per treatment (fifteen shrimp/replicate × three replicates) were randomly selected, fixed by liquid nitrogen at 6 h and 12 h after salinity challenge, and stored at −80 °C.

### 2.3. Salinity Acclimation Experiment of Post-Larvae Shrimp

Forty-five post-larvae shrimp (fifteen shrimp/bucket × three buckets) from seawater of 30‰ salinity were first sampled and set as the control group. Then, 1200 shrimp were randomized equally into 6 PVC buckets (140 L) with the consistent culture conditions as described above. They were divided into high-salinity and low-salinity acclimation groups with each containing three replicates (200 individuals per replicate). The salinity of each group was changed daily by the addition of high-salinity artificial seawater or freshwater during the water exchange process at 07:00 am. The salinity of the high-salinity acclimation experimental group was increased daily by 5‰ until it reached 50‰ salinity and was maintained at 50‰ salinity for 7 days. The low-salinity treatment group was acclimated from 30‰ to 10‰ through daily 5‰ decrements, following a continuous acclimation at 10‰ salinity for 7 days. Forty-five shrimp per treatment (fifteen shrimp/replicate × three replicates) were sampled at 07:00 pm of the day that the salinity reached the target salinity. Then, another sampling was performed after the 7 days of acclimation.

### 2.4. Acute Salinity Exposure Experiment of Adult Shrimp

Nine shrimp at 25‰ salinity (three shrimp/bucket × three buckets) were randomly selected as the control group (0 h), and the tissues of gill, hepatopancreas, intestine, and muscle were dissected, quickly frozen in liquid nitrogen, and stored at −80 °C. Based on a previous report from Boyd and Fast (1992), the salinity concentrations of 10‰ and 40‰ were chosen for the acute salinity exposure. Adult shrimp were directly transferred from seawater of 30‰ salinity to 10‰ or 40‰ salinity and each treatment had three replicates

(20 shrimp per replicate). Nine shrimp per treatment (three shrimp/replicate × 3 replicates) were chosen to fix the tissues as described above at 6 h and 12 h after salinity stress.

### 2.5. Salinity Acclimation Experiment of Adult Shrimp

Nine shrimp at 25‰ salinity (three shrimp/bucket × three buckets) were set as the control group and sampled as described in Section 2.4. Then, 180 shrimp in 6 buckets were divided into high-salinity and low-salinity acclimation groups with each containing 3 replicates (30 individuals per replicate). The two treatment groups were separately acclimated to 55‰ salinity or 10‰ salinity through daily 5‰ increments or decrements by using methods similar to those described in Section 2.3. Then, the sampling interval was set at 15‰ salinity and 9 shrimp per treatment (3 shrimp/replicate × 3 replicates) were sampled 12 h after reaching the salinity concentrations of 10‰, 40‰, and 55‰. Except for the sampled shrimp, few shrimp died during the acclimation period. In addition, no apparent changes in behavior, activity, and food intake were observed for the shrimp exposed to different salinities.

### 2.6. RNA Isolation and cDNA Synthesis

Total RNA of five post-larvae shrimp per replicate was extracted together using a Quick-RNA Microprep kit (Jianshi Bio, Beijing, China), and the total RNA of sampled tissues from adult shrimp was individually isolated by TRIzol Reagent (Vazyme, Nanjing, China). Thus, three RNA samples were obtained for each replicate. The RNA integrity was detected by 1% agarose gel electrophoresis. The purity and concentration of RNA were measured by NanoDrop 2000 spectrophotometer (Thermo Fisher Scientific, Waltham, MA, USA). Then, the first-strand cDNA was generated from an RNA sample of 1 μg with PrimeScriptTM RT Reagent Kit with gDNA Eraser (Vazyme, Nanjing, China).

### 2.7. Identification and Characterization of Osmoregulatory Genes

The mRNA sequences of *AQP3*, *AQP4*, *AQP11*, *CAc*, *NKAα*, and *VHA-G* from *L. vannamei* were used as query sequences to conduct online BLASTN (http://www.ncbi.nlm.nih.gov/BLAST/, accessed on 11 May 2021) against the TSA database of *M. japonicus* accessed on 11 May 2021. Transcripts homologous to the above genes were identified and specific primers were designed to amplify and validate the open reading frames (ORFs) of the transcripts (Table 1). Multiple sequence alignment of the obtained genes was conducted using the MUSCLE method in MEGA X software and visualized through Jalview software [33,34]. Phylogenetic analysis of these genes was performed by the neighbor-joining (NJ) method with 1000 bootstrap values in MEGA X software. Then, the phylogenetic trees were visualized using Interactive Tree of Life (iTOL) version 6 [35]. Transmembrane regions of NKAα protein were predicted with TMHMM 2.0 (http://www.cbs.dtu.dk/services/TMHMM, accessed on 11 May 2021).

**Table 1.** Specific primers used for gene cloning.

| Gene Name | Primer Type | Sequence (5′-3′) |
|---|---|---|
| *MjAQP3* | Forward | AGTGTACAGGAACAAAGGCTTAC |
| | Reverse | ACTACTCCTCCTTAGTCGCTCTC |
| *MjAQP4* | Forward | GAAGAGCAAGAAACCACCATC |
| | Reverse | CCGAGCCAGTAGATCTTTCAT |
| *MjAQP11* | Forward | CATCTCCGCAACCATGTCGAT |
| | Reverse | TGTGACACTACCGAAAATCCTC |
| *MjNKAα* | Forward | GCCTAGTGGCTTGTACATAAGTG |
| | Reverse | TGAGCTCTAGTAATTCTTGCGC |
| *MjCAc* | Forward | GACAGTTAAACATGGTTGGCTG |
| | Reverse | GAGTGTCACTTATTTCTCTGGTCA |
| *MjVHA-G* | Forward | ACAAGACACACACAGCAGACG |
| | Reverse | CAGTTCTACACACTTATCATGGAGT |

### 2.8. Expression Analysis of Osmoregulatory Genes

Specific primers of target genes and reference gene (elongation factor 1 alpha, EF1$\alpha$) were designed for the tissue distribution of the target genes and their expression patterns under salinity stresses (Table 2). The qPCR reaction mixture contained 5.0 µL of 2 × ChamQ Universal SYBR qPCR Master Mix (Vazyme, Nanjing, China), 2.0 µL of cDNA (5 ng/µL), 0.2 µL each of 10 µM forward and reverse primers, and 2.6 µL of RNase-free water. The entire reaction was carried out in 96-well PCR plates by a CFX96 Touch Real-Time PCR Detection System (Bio-Rad, Hercules, CA, USA). The qPCR amplification process consisted of pre-incubation at 95 °C (30 s) followed by 40 cycles of 95 °C (10 s) and 60 °C (30 s). Standard melt curve analysis was performed to confirm the amplification of individual qPCR products. The accumulation of fluorescence signals from the SYBR Green dye was recorded in the 60 °C (30 s) phase during each cycle. A negative control (no-template reaction) was included throughout. Each sample was analyzed in triplicates. The relative gene expression levels were calculated by the comparative Ct method using the formula $2^{-\Delta\Delta Ct}$ (Livak and Schmittgen, 2001). Statistical analysis of the qPCR results was evaluated by one-way analysis of variance (ANOVA) or Two-way ANOVA-ordinary, followed by Tukey's test embedded in Graphpad Prism 8.0 (GraphPad Software, San Diego, CA, USA), wherein $p < 0.05$ denoted a statistically significant difference.

**Table 2.** Specific primers used for qPCR analysis.

| Gene Name | Primer Type | Sequence (5′-3′) | Melting Temperature | Efficiency |
|---|---|---|---|---|
| *MjAQP3* | Forward<br>Reverse | ACCTTGGGTGTGCTCGTATCT<br>TGGGCAATAACATACACGGG | 86.0 °C | 97.23% |
| *MjAQP4* | Forward<br>Reverse | TGGCTGCTGCTCTTATCTACTCC<br>CGTTGTTGGTTCGCTTATTGTA | 81.0 °C | 95.36% |
| *MjAQP11* | Forward<br>Reverse | CGACCTCCAAGTGCCTGTAG<br>AAGCACGGGATTGAAGTAACC | 85.0 °C | 96.53% |
| *MjNKA$\alpha$* | Forward<br>Reverse | ACCCGCCGTAACTCTATTGTC<br>TGCCTGGGGTGTAGGAAAG | 82.5 °C | 96.97% |
| *MjCAc* | Forward<br>Reverse | ACATTCATGGAAGGCTCAGGT<br>AGTAGCAGCGACCATTGACG | 84.0 °C | 95.67% |
| *MjVHA-G* | Forward<br>Reverse | TGAATAAGCAGGTTGCCCAC<br>CCTTCTTAGCGTTGGTGTGC | 81.5 °C | 96.78% |
| *EF1$\alpha$* | Forward<br>Reverse | GGAACTGGAGGCAGGACC<br>AGCCACCGTTTGCTTCAT | 85.0 °C | 98.98% |

## 3. Results

### 3.1. Molecular Characterization of Osmoregulatory Genes

Three transcripts from the *M. japonicus* TSA database were identified to encode three types of AQPs proteins. The ORF sequences were further verified by PCR amplification and Sanger sequencing. The mRNA sequences were submitted to GenBank with the accession numbers OP823702, OP823703, and OP823704. The ORF of *MjAQP3*, *MjAQP4*, and *MjAQP11* consisted of 978, 786, and 780 nucleotides, encoding 325, 261, and 259 amino acids, respectively (Figure S1). Multiple sequence alignment of MjAQPs with AQPs of other animals showed that MjAQP3 and MjAQP4 had conserved NPA motifs, while MjAQP11 had non-canonical cysteine-proline-tyrosine (CPY) and asparagine-proline-valine (NPV) motifs (Figures S1 and S2). Phylogenetic analysis showed that they were closely clustered with their homologous proteins from *L. vannamei* and separately belonged to the groups of aquaglyceroporins, the classical aquaporins, and the unorthodox aquaporins (Figure 1A). Therefore, the MjAQPs were named MjAQP3, MjAQP4, and MjAQP11 based on their homologous genes in *L. vannamei*.

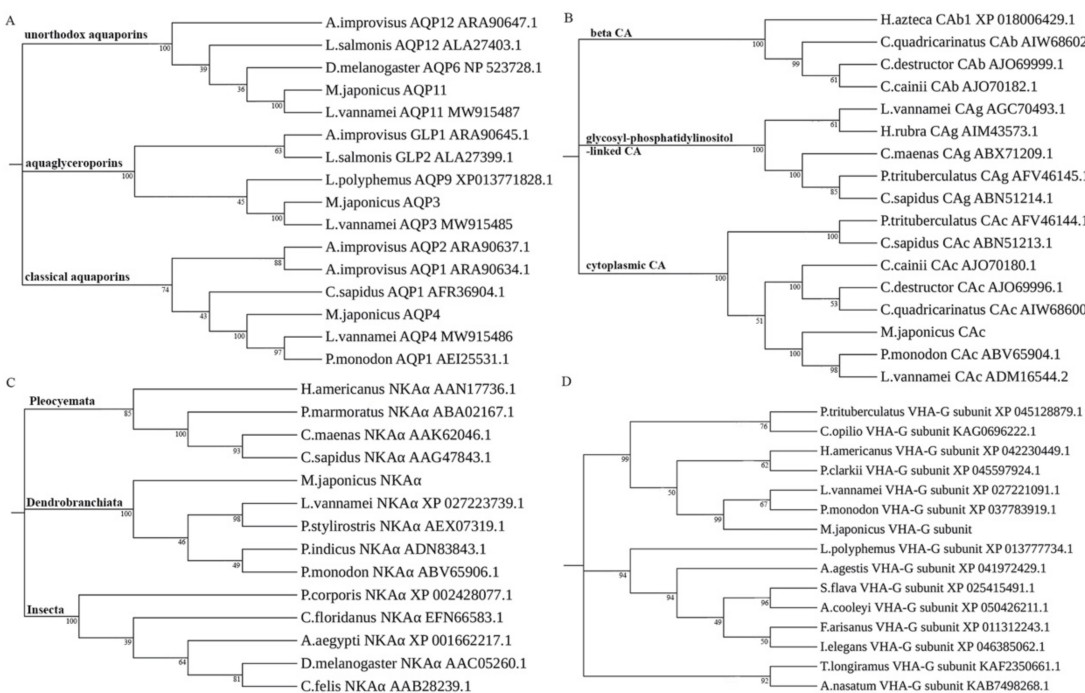

**Figure 1.** Phylogenetic analysis of (**A**) MjAQPs, (**B**) MjCAc, (**C**) MjNKAα, and (**D**) MjVHA-G based on NJ method. Phylogenetic trees based on the amino acid sequences of the AQPs, CA, NKAα, and VHA-G in various species were constructed by using MUSCLE alignment and the NJ method through MEGA X software. The node value indicates the percentage of 1000 bootstrap replicates.

Transcripts separately encoding the NKA α subunit (NKAα), cytoplasmic CA (CAc), and VHA G subunit (VHA-G) were also confirmed, and the ORF and deduced amino acids sequences are shown in Figure S2. The mRNA sequences were submitted to GenBank with the accession numbers OP823705, OP823706, and OP823707. The ORF of *MjNKAα*, *MjCAc*, and *MjVHA-G* consisted of 3039, 813, and 357 nucleotides, encoding 1012, 270, and 118 amino acids, respectively (Figure S3). The deduced MjCAc contained an alpha carbonic anhydrase signature, a proton acceptor site, and three zinc-binding histidine residues (Figures S3A and S4). Phylogenetic analysis revealed that the CA proteins were classified into three sub-groups and the deduced MjCAc was clustered with the CAc proteins from decapod crustaceans (Figure 1B). In addition, the TMHMM prediction indicated that the MjNKAα contained eight trans-membrane helices and the trans-membrane domains were highly conserved among decapod NKAα proteins (Figures S3B and S5). Based on the phylogenetic analysis, the decapod NKAα proteins were grouped into two branches (Pleocyemata and Dendrobranchiata) and were distinct from the insect NKAα proteins. MjNKAα belonged to the Dendrobranchiata sub-group and was clustered with NKAα proteins from penaeid shrimp (Figure 1C). Moreover, multiple sequence alignments showed that MjVHA-G shared high similarities with the decapod VHA-G proteins and MjVHA-G was clustered with the VHA-G proteins from penaeid shrimp in the phylogenetic tree (Figure 1D and Figure S6)

### 3.2. Tissue Expression Patterns of Genes in Adult Shrimp

Tissue expression analysis revealed the differential expression patterns of the *MjAQPs* genes across the osmoregulatory tissues. *MjAQP3* was expressed at the highest level in the intestine, while it was not detected in the gills (Figure 2A). In addition, the *MjAQP3* gene was the most highly expressed aquaporin gene in the intestine (Figure S7). Although the highest expression levels of *MjAQP4* and *MjAQP11* were both observed in the muscle, the *MjAQP11* gene was much more highly expressed than *MjAQP4* in the muscle

(Figure 2A and Figure S7). On the contrary, the expression level of *MjAQP4* was higher than those of other *MjAQPs* in the gill (Figure S7).

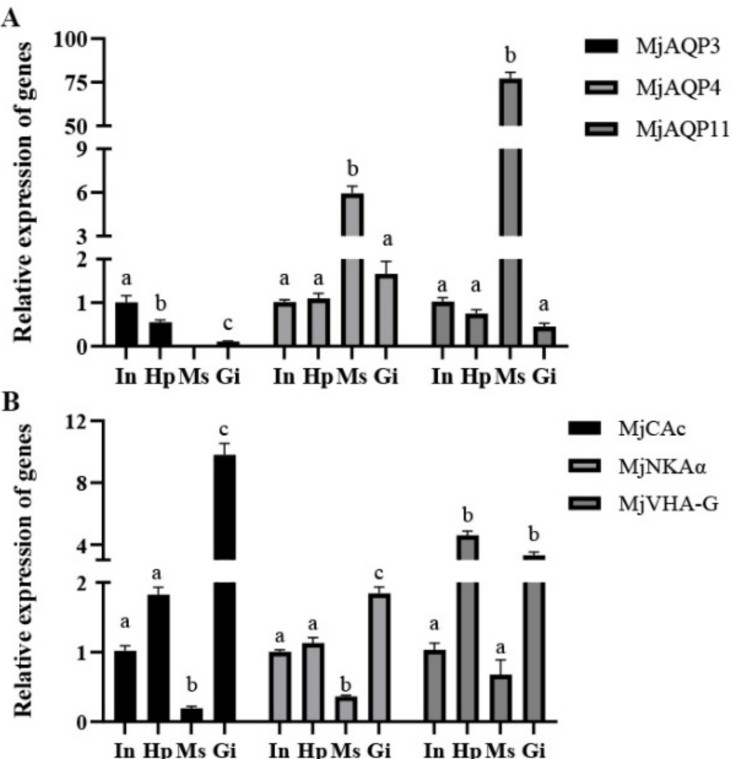

**Figure 2.** The relative expression levels of (**A**) *MjAQPs* and (**B**) *MjCAc*, *MjNKAα*, and *MjVHA-G* in different tissues of *M. japonicus*. Abbreviations: In, intestine; Hp, hepatopancreas; Ms, muscle; Gi, gill. The relative expression level of each gene in the intestine is separately used as calibrator. The data are presented as mean + SD ($n$ = 3). Means labeled with different letters are significantly different ($p < 0.05$, one-way ANOVA, Tukey).

The tissue distribution patterns of the ion-transportation-related genes were relatively uniform among the osmoregulatory tissues. (Figure 2B). The *MjCAc* expression profile was similar to that of the *MjNKAα* gene with the highest levels in the gills and the lowest levels in the muscle, while both genes showed intermediate expression in the hepatopancreas and muscle. As to the expression of *MjVHA-G*, it was highly expressed in the gills and hepatopancreas and lowly expressed in the muscle and intestine.

### 3.3. Effects of Acute Salinity Stress on Post-Larvae Shrimp

It was noted that with the declining salinity, the mortality of post-larvae shrimp increased over time (Figure 3A). More than half of the post-larvae shrimp could survive for 12 h when directly exposed to 10‰ salinity, while all died under 5‰ salinity after exposure for 12 h. As the salinity stresses continued for 24 h, the mortality sharply elevated at 10‰ and 15‰ salinity. Similar trends were observed for mortality with the rising salinity (Figure 3B). In addition, the majority of the post-larvae shrimp could survive for 12 h when directly exposed to 50‰ salinity, while all died under 55‰ salinity after exposure for 12 h. Moreover, the mortality rate of the shrimp at 50‰ salinity increased slightly during the following 12 h, showing a good tolerance to high-salinity stress. Based on the performance of post-larvae shrimp at different salinity concentrations, exposure to 10‰ and 50‰ salinity for 12 h were chosen for the following acute salinity stresses.

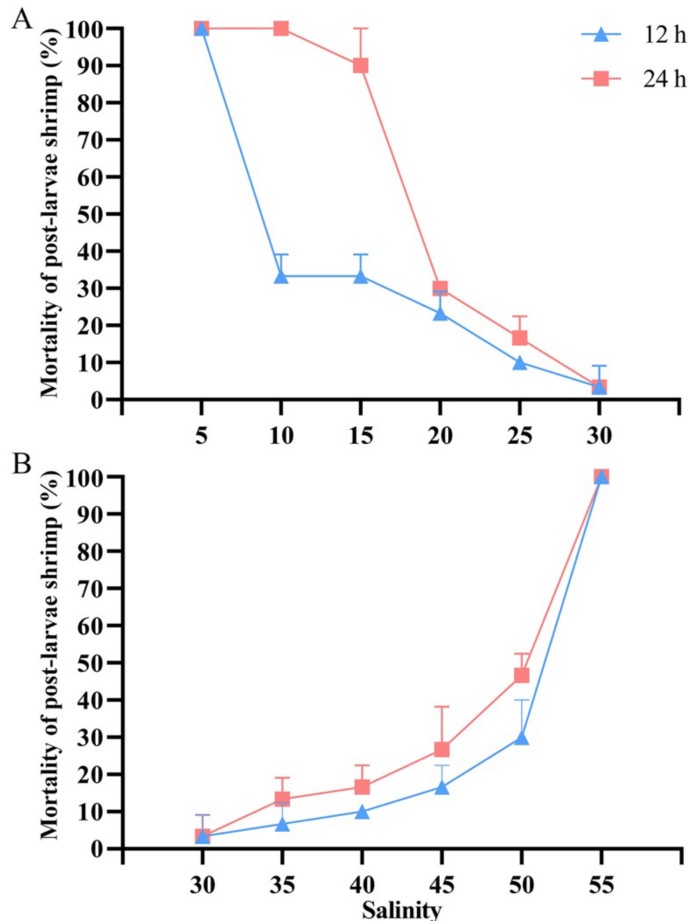

**Figure 3.** Cumulative mortality rates of post-larvae shrimp under acute (**A**) low-salinity and (**B**) high-salinity stress. Each bar represents the mean value from three repetitions with SD.

The expression level of *MjAQP3* was significantly decreased at 6 h and then remained unchanged from 6 h to 12 h under 10‰ salinity (Figure 4A). In addition, the expression of *MjAQP3* was also significantly down-regulated at 6 h, up-regulated from 6 h to 12 h, and still significantly lower than the initial level at the salinity of 50‰. Nevertheless, the expression level of *MjAQP3* at 50‰ salinity was significantly higher than that at 10‰ salinity after 12 h of exposure (Figure 4A). There was a progressive increase in the expression levels of *MjAQP4* from 0 h to 12 h at 10‰ salinity (Figure 4B). The expression levels of *MjAQP4* were not significantly changed at 6 h, and then significantly increased from 6 h to 12 h at 50‰ salinity (Figure 4B). Moreover, there was a significant decline in the expression levels of *MjAQP11* from 0 h to 12 h at both 10‰ and 50‰ salinity (Figure 4C).

A progressive decline in the expression level of *MjNKAα* was found from 0 h to 12 h at 10‰ salinity, whereas the expression of *MjNKAα* was significantly down-regulated at 6 h, then up-regulated from 6 h to 12 h, and was significantly higher than the initial level at the salinity of 50‰ (Figure 4D). The expression levels of *MjCAc* were significantly decreased at 6 h and then remained unchanged from 6 h to 12 h at both 10‰ and 50‰ salinity concentrations (Figure 4E). Additionally, a significantly elevated expression of *MjVHA-G* was observed at 6 h and the expression did not change significantly from 6 h to 12 h at 10‰ salinity. The expression of *MjVHA-G* was not significantly changed at 6 h, and then significantly increased from 6 h to 12 h at 50‰ salinity (Figure 4F).

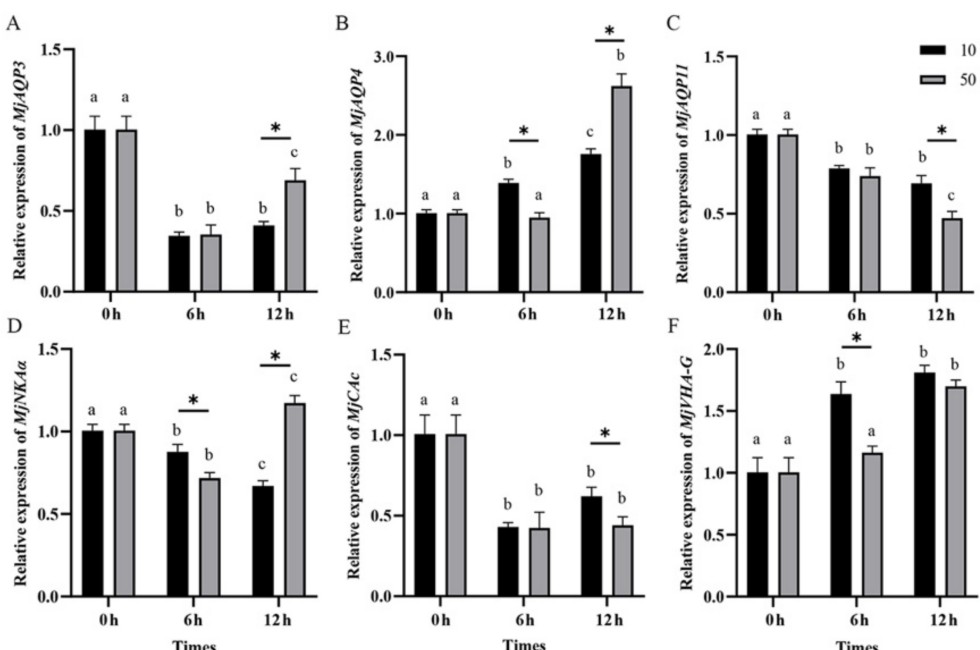

**Figure 4.** The relative expression levels of (**A**) *MjAQP3*, (**B**) *MjAQP4*, (**C**) *MjAQP11*, (**D**) *MjNKAα*, (**E**) *MjCAc*, and (**F**) *MjVHA-G* in the post-larvae shrimp at 10‰ and 50‰ salinity stresses. The relative expression level of each gene at 0 h is separately used as calibrator. The data are presented as mean + SD (*n* = 3). Lowercase letters indicate significant differences (*p* < 0.05) with the same salinity at different exposure times. Asterisks above black bars denote significant differences (*p* < 0.05) among different salinities at the same treatment time.

### 3.4. Gene Expression in Post-Larvae Shrimp during the Salinity Acclimation

Four types of expression patterns of the genes were found during the acclimation to 10‰ salinity (Figure 5A). The expression levels of *MjAQP3*, *MjAQP11*, and *MjNKAα* were significantly decreased when the salinity declined to 10‰ salinity and remained constant in the following acclimation period. Additionally, the expression of *MjAQP4* was slightly but not significantly elevated upon reaching 10‰ salinity, whereas its level was significantly higher than the initial value at the end of the acclimation period. Moreover, although the changing trend in the expression level of *MjCAc* was opposite to that noted for *MjVHA-G* when they acclimated to 10‰ salinity, both expression levels returned to the initial values after 7 d of long-term acclimation at 10‰ salinity.

These genes responded diversely to the high-salinity acclimation experiment (Figure 5B). There was no significant change in the expression of *MjAQP3* during the whole acclimation period. The expression of *MjAQP4* increased significantly and was maintained at a higher level, while the expression level of *MjAQP11* was significantly decreased at 50‰ salinity, followed by a further reduction after 7 d of acclimation. Moreover, the expressions of *MjCAc* and *MjNKAα* first significantly declined at 50‰ salinity, and then returned to the initial level during further acclimation. *MjVHA-G* expression was slightly but not significantly elevated at 50‰ salinity, whereas its level was significantly increased during the following acclimation.

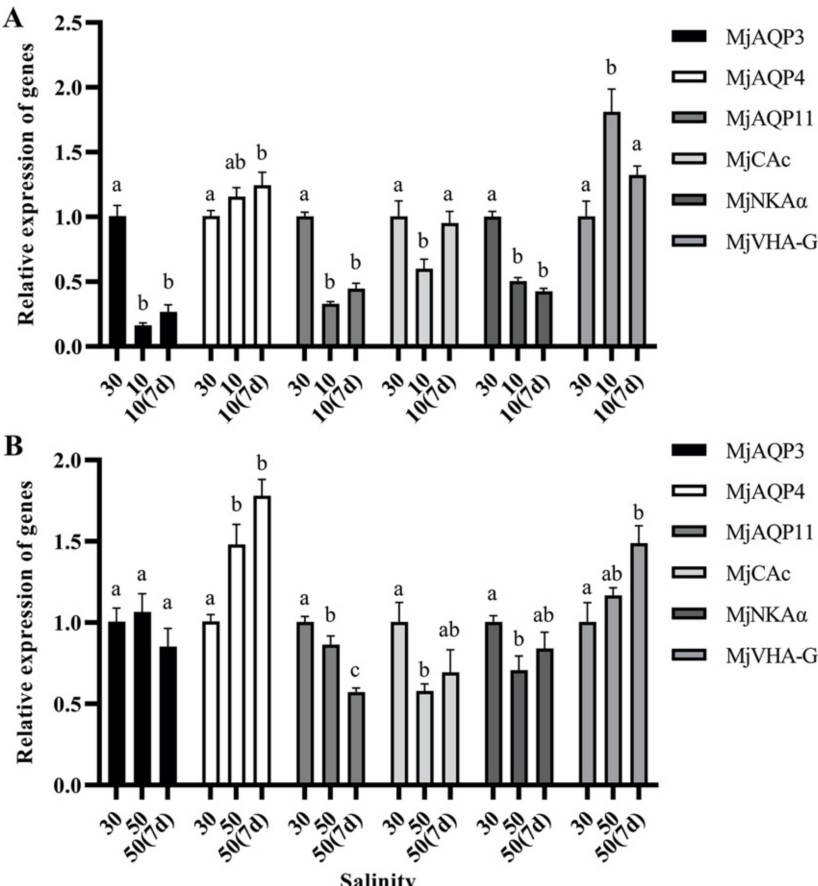

**Figure 5.** The relative expression levels of genes during the low-salinity (**A**) and high-salinity (**B**) acclimation periods in the post-larvae shrimp. The relative expression level of each gene at 30‰ salinity is separately used as calibrator. The data are presented as mean + SD (*n* = 3). Lowercase letters indicate significant differences (*p* < 0.05) with each gene.

### 3.5. Gene Expression in Response to Acute Salinity Stresses in the Gills

There was a significant increase in the expression of *MjAQP3* and *MjAQP11* after 6 h of acute low-salinity and high-salinity stress (Figure 6A,C). Then, *MjAQP3* expression declined and was still higher than the initial level, while *MjAQP11* expression decreased to the initial value from 6 h to 12 h. Moreover, *MjAQP4's* expression level at 40‰ salinity did not vary significantly up to 12 h, while its expression level at 10‰ salinity increased significantly at 6 h and then remained high from 6 h to 12 h (Figure 6B).

The expression of *MjNKAα* at 40‰ salinity showed a trend of rising first and then falling to the initial level, while it increased significantly from 6 h to 12 h at 10‰ salinity (Figure 6D). In addition, the *MjCAc* expression level at 40‰ did not vary significantly up to 12 h. Nevertheless, its expression level at 10‰ increased significantly from 6 h to 12 h and was significantly higher than that at 10‰ after 12 h of salinity exposure (Figure 6E). *MjVHA-G* exhibited the opposite response to low-salinity and high-salinity stress. The expression of *MjVHA-G* at 40‰ salinity decreased significantly at 6 h and then remained low from 6 h to 12 h, while it increased significantly from 6 h to 12 h at 10‰ salinity (Figure 6F).

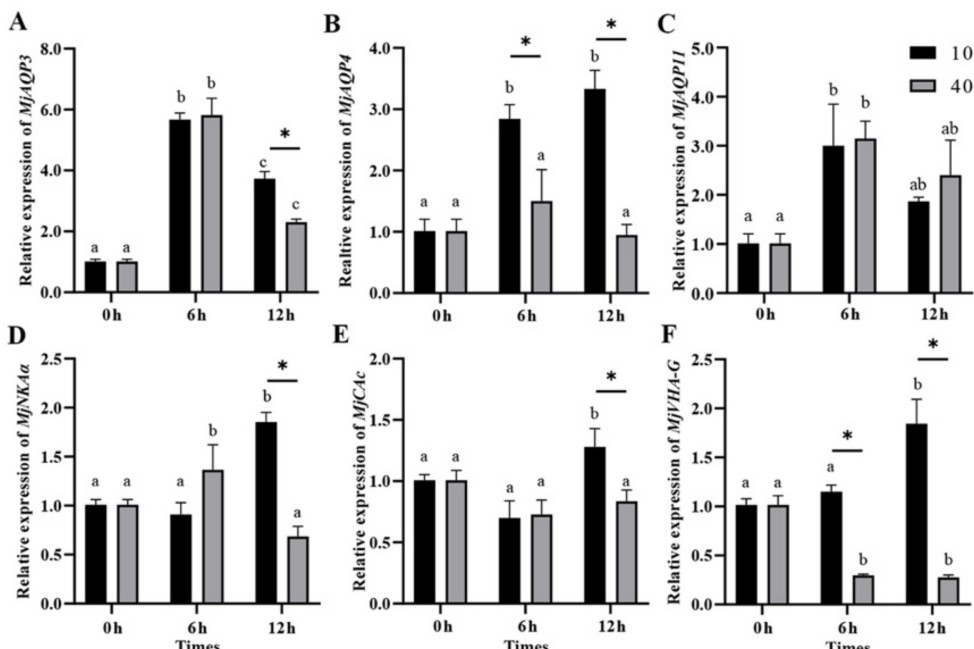

**Figure 6.** The relative expression levels of (**A**) *MjAQP3*, (**B**) *MjAQP4*, (**C**) *MjAQP11*, (**D**) *MjNKAα*, (**E**) *MjCAc*, and (**F**) *MjVHA-G* in the gills at 10‰ and 40‰ salinity stresses. The relative expression level of each gene at 0 h is separately used as calibrator. The data are presented as mean + SD ($n = 3$). Lowercase letters indicate significant differences ($p < 0.05$) with the same salinity at different exposure times. Asterisks above black bars denote significant differences ($p < 0.05$) among different salinities at the same treatment time.

### 3.6. Gene Expression in Response to Salinity Acclimation in the Gills

There was a significant increase in the expression of *MjAQP3*, *MjAQP11*, and *MjNKAα* concomitant with a significant decrease in the expression of *MjCAc* in the gills, whereas the expression of *MjAQP4* and *MjVHA-G* remained unchanged when adult shrimp were acclimated from 25‰ to 10‰ salinity (Figure 7A). During the high-salinity acclimation period, the expression of *MjAQP4* and *MjVHA-G* significantly declined at 40‰ salinity and was maintained at low levels at up to 55‰ salinity (Figure 7B). The expression levels of *MjAQP3* and *MjCAc* were not significantly changed at 40‰ salinity, while they were significantly decreased at 55‰ salinity. In contrast, the expression of *MjAQP11* was significantly elevated when it acclimated from 40‰ salinity to 55‰. Aside from the latter, there was no significant difference in the expression level of *MjNKAα* across the whole high-salinity acclimation period (Figure 7B).

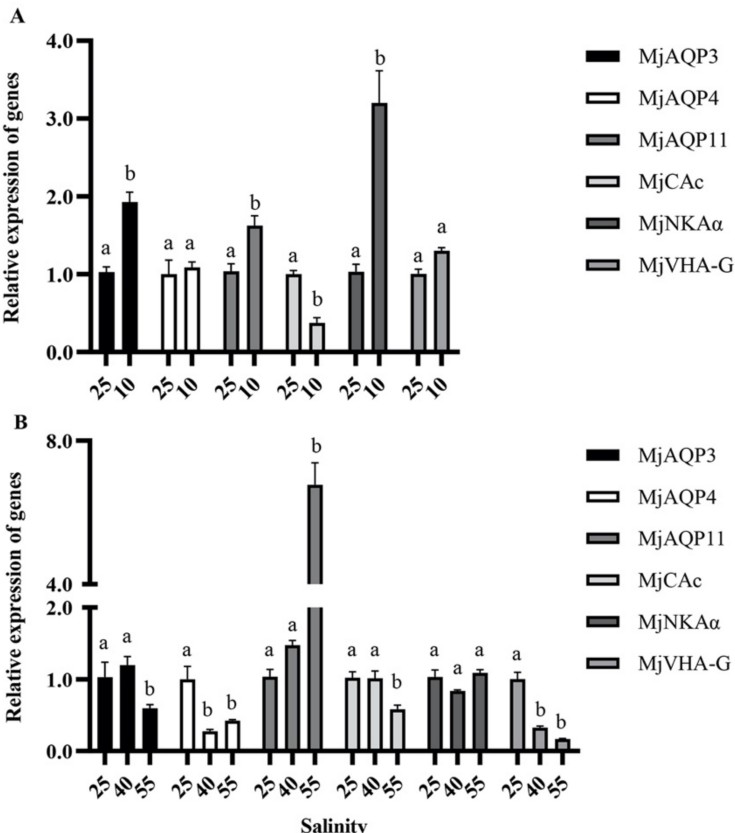

**Figure 7.** The relative expression levels of genes during the low-salinity (**A**) and high-salinity (**B**) acclimation periods in the gills. The relative expression level of each gene at 25‰ salinity is separately used as calibrator. The data are presented as mean + SD ($n$ = 3). Lowercase letters indicate significant differences ($p$ < 0.05) with each gene.

## 4. Discussion

### 4.1. Tolerance of M. japonicus to Salinity Challenges

It has been reported that the kuruma shrimp can tolerate a wide range of salinities from 10‰ and 40‰ [32]. In this study, the post-larvae shrimp could not survive under the acute stress at 5‰ salinity for 12 h, suggesting that 5‰ salinity is lethal for *M. japonicus*. Recently, juvenile kuruma shrimp were reported to survive at 8‰ salinity for 96 h [36]. These results further confirm that the lower limit of salinity for *M. japonicus* to survive is ~10‰ salinity. Minor differences may result from the differences in the developmental stages of shrimp. Nevertheless, the intriguing findings are that *M. japonicus* could tolerate salinity up to 50‰ for the post-larvae shrimp and 55‰ for the adult shrimp during the acclimation process. In previous studies, salinities of 40‰ or 42‰ were commonly used as the salinity of acclimation in *M. japonicus* [37–40]. The current study is the first to report that the kuruma shrimp can be acclimated to 55‰ salinity. As the salinity of the high-salinity waters in northern China is more than 40‰ [30], *M. japonicus* has significant potential for aquaculture in these areas. The post-larvae shrimp could be first acclimated to the target salinity and then cultivated in outdoor ponds with high-salinity seawaters. Further studies are needed to comprehensively evaluate the growth, survival rate, and disease resistance of *M. japonicus* during long-term cultivation in high-salinity seawaters.

*4.2. Expression of MjAQPs in Response to Salinity Challenges*

Depending on the tertiary structure and selective permeability, vertebrate AQPs are classified into three categories: the classical aquaporins (AQP0, 1, 2, 4, 5, 6, and 8) for water transportation, the aquaglyceroporins (AQP3, 7, 9, and 10) potentially transporting water, and other small solutes, and the unorthodox aquaporins (AQP11 and 12) with an undetermined function [17,18]. However, although the three sub-families of AQPs have also been identified in crustaceans, the members of each sub-family differ significantly from those in vertebrates. A previous study has reported the identification and characterization of unique AQPs for each sub-type of *L. vannamei* [26]. In this study, three MjAQPs were found through an online BLAST search against AQPs from *L. vannamei*, which separately belong to the groups of aquaglyceroporins, classical aquaporins, and unorthodox aquaporins.

The *MjAQP3* mRNA level in the post-larvae shrimp at 30‰ salinity was higher than that under acute low-salinity stress or long-term acclimation to low salinity. Similarly, the expression level of an aquaglyceroporin, *CmGLP1*, was decreased in the green crab (*Carcinus maenas*) under low-salinity exposure in comparison to crabs at normal seawater salinity [41]. Moreover, the expression of *MjAQP3* in the post-larvae shrimp at 30‰ salinity was greater than that under acute high-salinity stress. In addition, the expression of *AQP3* was also down-regulated in the intestine under acute high-salinity challenges in *L. vannamei* [26]. Besides water, the aquaglyceroporins also transport glycerol, urea, and other small solutes [16,42]. Moreover, *MjAQP3* is primarily expressed in the intestine and hepatopancreas. The MjAQP3 gene may be involved in not only water transportation but also glycerol transportation for nutrient absorption along the intestine and hepatopancreas. The decreased expression of *MjAQP3* implies weak functions regarding the digestion and absorption of nutrition under salinity stress. Aside from *MjAQP3*, the expression levels of MjAQP11 were also decreased in the post-larvae shrimp when exposed to low-salinity or high-salinity stress. Similar expression patterns were found for *AQP11* in the muscle of *L. vannamei* [26]. Although the unorthodox AQPs have been reported to transport water and glycerol in the oocytes of medaka and zebrafish [43,44], there is still a lack of understanding regarding how unorthodox aquaporins regulate osmotic pressure in crustaceans. Nevertheless, these results imply the response of unorthodox aquaporins to the salinity changes in the shrimp. In contrast to the expression profiles of *MjAQP3* and *MjAQP11*, the expression of *MjAQP4* was up-regulated in the post-larvae shrimp under acute and long-term salinity challenges. The expression of a classical aquaporin, *CsAQP1*, is increased in larval blue crabs (*Callinectes sapidus*) after being exposed to low-salinity stress for 96 h [45]. In addition, *AQP4* expression is elevated in the hepatopancreas of *L. vannamei* under acute high-salinity stress [46]. As the unique classical AQP in the shrimp, MjAQP4 should be primarily for transporting water and regulating cell volume, and thus has a critical role in the salinity adaption of the post-larvae shrimp.

In the post-larvae shrimp, we could only detect changes in the global expression levels of *MjAQPs*. However, a tissue distribution analysis implied that the *MjAQPs* have distinct tissue specificity, suggesting their different functions in different tissues. A higher expression of *MjAQP3* and *MjAQP11* was found in the gills under salinity stresses. Elevated expression levels of *AQP3* have been reported in the gills of teleost fish in a high-salinity environment [43,47–51]. However, the results contradict their expression trends in the post-larvae shrimp. These inconsistent expression profiles may be related to the lower expression levels of *MjAQP3* and *MjAQP11* in the gills. Thus, the increased expression levels in the gills could not counteract the declined levels in other tissues. Therefore, it is necessary to further analyze the *MjAQPs'* expression in one tissue to elucidate the roles of *MjAQPs* in adaption to salinity stress.

As the primary organ for osmoregulation, the gills sense salinity changes in their surrounding environment and adapt through the modulation of gene expression. The expression of *MjAQP4* was not significantly changed in the gills under acute high-salinity stress, while it was significantly increased in the gills under acute low-salinity stress. Moreover, the expression of *MjAQP3* and *MjAQP11* was elevated in both treatments. As the

most prominently expressed *AQP* gene in the gills, *MjAQP4* is likely to play a dominant role in water transport. Accordingly, the *MjAQP3* and *MjAQP11* genes may have an assisting role in water transport. The results suggest that the global expression of *MjAQPs* may be orchestrated by the extent of the salinity shift. In this study, the kuruma shrimp show better tolerance to high-salinity stress. Thus, we conclude that low-salinity stress may present greater osmotic stress for the gills. Therefore, the expression of *MjAQPs* quickly increased under acute low-salinity stress, whereas only the up-regulation of *MjAQP3* and *MjAQP11* is required under acute high-salinity stress. The gills face the challenges of water influx in hypoosmotic conditions and the loss of water under hyperosmotic conditions [20,52,53]. Osmotic pressure from salinity changes drives water flow across the plasma membrane through *AQPs*, while the AQP channel determines the speed but not the direction of water flow [54]. The increased expression of *AQPs* is needed to excrete the excess water of the cells or to gain more water from the ambient environment and maintain the balance of intracellular volume and osmotic pressure. The expression of *AQP* is also higher at 0‰ and 10‰ salinity treatments than at 5‰ salinity in the gills of the prawn *Macrobrachium australiense* [23].

Challenges to the gills during the salinity acclimation process are different from those under acute salinity stress. Thus, the *MjAQPs* showed differential patterns in the gills in response to the salinity acclimation challenge. The salinity decreased by 5‰ daily during the low-salinity acclimation period and such minor salinity perturbations may cause reduced osmotic stress on the gills. This situation is similar to that for the gills under acute high salinity. Thus, the expression of *MjAQP4* was not significantly changed, and the expression of *MjAQP3* and *MjAQP11* was elevated in the gills during the acclimation to 10‰ salinity. Moreover, the continuous high-salinity stress possibly imposes pressure on excessive water loss. To limit the rate at which water equilibrated across tissues during high-salinity acclimation, *MjAQP4* expression significantly decreased at 40‰ and 55‰ salinity, leading to reductions in water transport in the gills.

### 4.3. Expression of Ion-Transportation-Related Genes in Response to Salinity Challenges

NKA is thought to be the main force driving the trans-epithelial movement of monovalent ions across the gill epithelia in crustaceans [7]. It consists of a single α-subunit, a glycosylated β-subunit, and a γ-subunit, of which the α-subunit contains an ATP-binding site, a phosphorylation site, and the amino acids essential for the binding of sodium and potassium ions [55]. Thus, NAKα has been shown to play an important role in osmoregulation in aquatic organisms under salinity challenges. In this study, the ORF sequence of NKAα was cloned from *M. japonicus* and a further phylogenetic analysis showed that MjNKA belonged to the shrimp NKAα sub-group. Previous studies have reported the enhanced expression of *NKAα* in the gills of euryhaline crustaceans subjected to osmotic stress [56–60]. The expression of *MjNKAα* increased significantly in the gills of the shrimp exposed to 10‰ or that acclimated to 10‰ salinity. The higher expression of *MjNKAα* may function to maintain the osmolality, composition, and volume of the hemolymph by actively transporting and exchanging $Na^+$ and other critical cations through the gills. Nevertheless, its expression did not significantly change in the gills under high-salinity conditions. Significant changes in *NKAα* expression were not detected in the gills of swimming crabs (*Portunus trituberculatus*) under the low- and high-salinity conditions [22]. In addition, the NKA activity was unchanged in the gills of blue crabs (*Callinectes danae*) acclimated to 15‰ salinity and giant freshwater prawns (*Macrobrachium rosenbergii*) that acclimated from high-salinity to freshwater conditions [61,62]. These results suggest that the expression of *NKAα* might relate to the variable salinity tolerance among species. Beyond that, NKA activity has been shown to correlate with the developmental stages of *M. japonicas* [63], which may be the reason for the down-regulation of *MjNKAα* in the post-larvae shrimp in low-salinity conditions. Another reason may be that the expression of *MjNKAα* decreased in the other tissues of post-larvae shrimp. Recently, the expression of *MjNKAα* was found

to be down-regulated in the hepatopancreas of juvenile shrimp after 24 h of low-salinity stress [36].

CA catalyzes the reversible hydration reaction of carbon dioxide ($CO_2$) into protons ($H^+$) and bicarbonate ($HCO_3^-$) [64]. There are three types of CA in arthropods, including cytoplasmic CA (CAc), glycosyl-phosphatidylinositol-linked CA (CAg), and beta CA (CAb) [65]. The CAc isoform exists in the cytoplasm and provides counter ions ($H^+$ and $HCO_3^-$) for the exchange of $Na^+/H^+$ and $Cl^-/HCO_3^-$ across the plasma membrane; thus, it is directly involved in osmotic and ionic regulation [66]. In this study, the ORF sequence of one CA was cloned from *M. japonicus* and further phylogenetic analysis confirmed that the MjCA was clustered into the CAc sub-group. The expression of *CAc* typically increases rapidly to supply the counter ions used in NaCl uptake under low-salinity challenges in the gills of euryhaline crustaceans [14,67–71]. The expression of *MjCAc* in the gills also increased significantly after a 12 h exposure to low salinity. However, beyond that, the expression levels of *MjCAc* were significantly decreased in the post-larvae shrimp following acute low-salinity stress. Moreover, decreased *MjCAc* expression was also observed in the gills of shrimp after the acclimation to low salinity. *CA* has been reported to show a lower expression level in *P. trituberculatus* after exposure to a low salinity of 11 ppt [72]. The reduction in the expression of *CA* mRNA might be due to the extremely low-salinity conditions. In this study, 10‰ salinity nearly reaches the limit of salinity for the survival of *M. japonicus*. The activity of CA and expression of *CA* mRNA in the gills increased with the decreased salinity when the *C. maenas* subjects were transferred from 32 ppt salinity to 25, 20, and 15 ppt salinity, while both reduced when the salinity continued to drop to 10 ppt [67]. These results suggest that extreme hypoosmotic stress may impede the induction of *CA* mRNA. Furthermore, the expression of *MjCA* was also significantly decreased in the post-larvae shrimp or the gills when exposed to high-salinity conditions. The expression of *CA* was down-regulated in the gills of *P. trituberculatus* after their long-term adaption to high salinity [22]. In hyperosmotic conditions, organisms must prevent high concentrations of external ions from penetrating the body. Thus, the down-regulation of *MjCAc* may involve the active secretion of NaCl to compensate for the passive NaCl influx from the external environment.

VHA is a multiple-subunit protein containing two domains: $V_0$ and $V_1$. The $V_1$ domain consists of eight different subunits (A–H) responsible for ATP hydrolysis, and the $V_0$ domain is composed of six subunits and is involved in the translocation of protons across the membrane [73]. Thus, VHA can participate in ion regulation by actively transporting protons into the medium, which in turn allows $Na^+$ to flow in through Na channels and $Na^+/H^+$ exchangers [74,75]. A previous transcriptome analysis showed that the *VHA-G* subunit had the highest expression levels and was differentially expressed under salinity stress in the gills of *L. vannamei* [1]. Based on this result, we cloned the *VHA-G* subunit of *M. japonicus* to explore the function of *MjVHA-G* in response to salinity stress. The expression of *MjVHA-G* in the gills increased under low-salinity stress and was down-regulated following high-salinity stress. Similar expression of patterns of *VHA* subunit mRNA have been reported in crustaceans [4,76–78]. However, the expression of *MjVHA-G* in the post-larvae shrimp increased significantly at high-salinity conditions, which was not consistent with those in the gills. As it also shows high expression in the hepatopancreas, this inconformity may be due to its expression in other tissues.

Taken together, the expression levels of *MjNKAα*, *MjCAc*, and *MjVHA-G* were all promoted in the gills under acute low-salinity conditions. MjCA catalyzes the hydration of $CO_2$ to provide $H^+$ and $HCO_3^-$, and then MjVHA complements MjNAK by powering sodium and chloride ions' uptake by the excretion of $H^+$ and $HCO_3^-$ in lowly ionic environments. In addition, *MjCAc* and *MjVHA-G* are down-regulated in response to high salinity to reduce the intake of sodium and chloride ions by the epithelial cells in the gills.

## 5. Conclusions

It is essential for crustaceans to maintain ionic balance and water homeostasis to regulate osmoregulation in a salinity-varying environment. This study mainly investigated the expression patterns of key osmoregulatory genes involved in ion and water transportation in *M. japonicus* at different salinities. The results reveal remarkable differences in the patterns of gene expression depending on the intensity, exposure time, and treatment conditions of salinity stress. Overall, *MjAQPs* show varied tissue distribution patterns and are operative to different extents in different tissues and osmotic conditions. On the contrary, *MjNKAα*, *MjCAc*, and *MjVHA-G* have some consistent tissue distribution patterns and tend to act in tight cooperation to allow for effective osmoregulation.

**Supplementary Materials:** The following are available online at https://www.mdpi.com/article/10.3390/jmse10121870/s1, Figure S1: The ORFs and deduced amino acid sequences of (A) *MjAQP3*, (B) *MjAQP4*, and (C) *MjAQP11*. The asterisk stands for the stop codon. The NPA motifs are underlined with single line, Figure S2: Multiple alignment of deduced amino acid sequences of MjAQPs against selected AQPs sequences. Dark gray shading indicates residues conserved in >50% of the sequences, light gray shading indicates residues with similar properties. The NPA motifs are underlined with single line, Figure S3: The ORFs and deduced amino acid sequences of (A) *MjNKAα*, (B) *MjCAc*, and (C) *MjVHA-G*. The asterisk stands for the stop codon. The alpha-carbonic anhydrases signature is underlined. The zinc-binding histidine residues of MjCAc are in black box and the proton acceptor site of MjCAc is in black circle. The eight transmembrane domains of MjNKAα are underlined with single line, Figure S4: Multiple alignment of deduced amino acid sequences of MjCAc against selected CAc sequences. The conserved zinc-binding histidine residues are in red box and the most common proton acceptor sites are in blue box. Dark gray shading indicates residues conserved in >50% of the sequences, light gray shading indicates residues with similar properties, Figure S5: Multiple alignment of deduced amino acid sequences of MjNKAα against selected NKAα sequences. The eight transmembrane domains of NKAα are in red box. Dark gray shading indicates residues conserved in >50% of the sequences, light gray shading indicates residues with similar properties, Figure S6: Multiple alignment of deduced amino acid sequences of MjVHA-G against selected VHA-G sequences. Dark gray shading indicates residues conserved in >50% of the sequences, light gray shading indicates residues with similar properties, Figure S7: The relative expression levels of *MjAQPs* in different tissues of *M. japonicus*. Abbreviations: In, intestine; Hp, hepatopancreas; Ms, muscle; Gi, gill. The relative expression levels of *MjAQP3* in the intestine, hepatopancreas, and gill, and the relative expression level of *MjAQP4* in the muscle are separately used as calibrator. The data are presented as mean + SD ($n$ = 3). Means labeled with different letters are significantly different ($p$ < 0.05, one-way ANOVA, Tukey).

**Author Contributions:** Conceptualization, Y.C. and Y.L.; methodology, Z.Z.; software, Z.W.; validation, Y.L., F.L. and Y.C.; formal analysis, Z.Z. and L.Z.; investigation, R.Z. and X.W.; resources, Z.D. and J.S.; data curation, R.W.; writing—original draft preparation, Z.Z.; writing—review and editing, Y.C.; visualization, P.H.; supervision, Y.C.; project administration, Y.C.; funding acquisition, Y.C. and Y.L. All authors have read and agreed to the published version of the manuscript.

**Funding:** This work was supported by the Key Research and Development Plan in Shandong Province (2021LZGC027); the National Science Foundation of China (31802269; 31902407); the Shrimp and Crab Innovation Team of Shandong Agriculture Research System (SDAIT-15-011); the Natural Science Foundation of Shandong Province (ZR2019BC013); the High-level Talents Research Fund of Qingdao Agricultural University (663/1119054); and the "First Class Fishery Discipline" program in Shandong Province.

**Institutional Review Board Statement:** Not applicable.

**Informed Consent Statement:** Not applicable.

**Data Availability Statement:** The mRNA sequences in this study were submitted to GenBank with the accession numbers OP823702-OP823707.

**Conflicts of Interest:** The authors declare no conflict of interest.

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
