# Peer review of "How Do Gene Expression Patterns Change in Response to Osmotic Stresses in Kuruma Shrimp (Marsupenaeus japonicus)?"

_jmse, doi:10.3390/jmse10121870_

Round 1

Reviewer 1 Report

Is was with great pleaser that I read your manuscript. It is well presented, the methodology was very thorough, the results are well described, overall the work is fine, I could only fine minor corrections.

The topic presented is very relevant for shrimp aquaculture and presents new important related with the climate adaptation of these animals.

Reviewer 2 Report

The manuscript examined the expression of selected osmoregulatory genes in kuruma shrimp during salinity challlege and acclimation, both low and high salinity concentration. The design of the study, statistical analysis, and results interpretation were good. 

(1) L159-L163: Please explain the justification for additional acclimation experiments that extend the range of salinity further (up to 60% salinity and down to 5% salinity). Only the results at 55% salinity was presented. 

It seemed out of place in this section when (1) this part was done after the planned experiment is completed, (2) the range of salinity went beyond the originally planed concentrations, and (3) no results was given beyond the brief description. I think it might be more appropriate in the discussion section to point out the unexpected mortality as pitfalls and problems with your results. Then, give further information there. However, if this unexpected mortality is a major concern, a different rearing experiment (with ranges of salinity similar to that of PL shrimp) should have been performed and additional results provided in the paper. 

(2) Why there is no examination of the genes after 7 day acclimation for the adult stage (only on the day that a target salinity was reached)?  Please explain. 

(3) Why focus only on the gene expression of the gills in adult shrimp during salinity challenge/acclimation, when significant difference can be observed in other tissues (muscle/ hepatopancrease, Fig 2)? Please explain. 
